# Physical and Tactical Demands of the Goalkeeper in Football in Different Small-Sided Games

**DOI:** 10.3390/s19163605

**Published:** 2019-08-19

**Authors:** Daniel Jara, Enrique Ortega, Miguel-Ángel Gómez-Ruano, Matthias Weigelt, Brittany Nikolic, Pilar Sainz de Baranda

**Affiliations:** 1Department of Physical Activity and Sport, Faculty of Sport Science, Regional Campus of International Excellence “Campus Mare Nostrum”, University of Murcia, 30720 Murcia, Spain; 2Department of Sport Sciences, Polytechnic University of Madrid, 28031 Madrid, Spain; 3Department of Sport and Health, University of Paderborn, 33098 Paderborn, Germany; 4Department of Health, Exercise Science and Sport Management, University of Wisconsin-Parkside, Kenosha, WI 53141, USA

**Keywords:** sensors, data, small-sided games, methodology, training, sport

## Abstract

Background: Several studies have examined the differences between the different small-sided game (SSG) formats. However, only one study has analysed how the different variables that define SSGs can modify the goalkeeper’s behavior. The aim of the present study was to analyze how the modification of the pitch size in SSGs affects the physical demands of the goalkeepers. Methods: Three professional male football goalkeepers participated in this study. Three different SSG were analysed (62 m × 44 m for a large pitch; 50 m × 35 m for a medium pitch and 32 m × 23 m for a small pitch). Positional data of each goalkeeper was gathered using an 18.18 Hz global positioning system. The data gathered was used to compute players’ spatial exploration index, standard ellipse area, prediction ellipse area The distance covered, distance covered in different intensities and accelerations/decelerations were used to assess the players’ physical performance. Results and Conclusions: There were differences between small and large SSGs in relation to the distances covered at different intensities and pitch exploration. Intensities were lower when the pitch size was larger. Besides that, the pitch exploration variables increased along with the increment of the pitch size.

## 1. State of Art

Since their first application to field and team sports in 2006, global positioning systems (GPS) have been used as a technology to detect fatigue in matches, compare intensity profiles according to playing position, compare competition skill levels, and identify the most intense periods of play [1]. Besides the evaluation of (real) match performance, GPS technologies are today also frequently used to provide a more detailed assessment of the training load placed on players [2,3], because GPS devices provide a better understanding of the players’ individual training load by enabling detailed data to be collected, such as the movement speed and the distance covered during different drills and workouts [4]. The latest studies have implemented GPS data in their methodologies in order to specify athletes’ activity profiles [5], analyse specific game tasks [6], and/or define different tactical aspects [7], in order to better understand and optimize the game [8], prepare players to undertake the different positional match demands [9], specific physical and technical football drills [10], and for training tactical aspects [11], being used to enhance training content and subsequently improve performance [1]. Using GPS data, coaches can design physical conditioning and plan appropriate recovery time [12], following intense workouts according to the demands of each player’s position [13]. Thus, the quantification of different aspects of training load is crucial to understand the training process. In this sense, GPS data cannot only be used to analyze game load and match performance during competition, but also during training [10,14]. As the GPS technology continues to develop, it helps coaches to determine the appropriate training load, to improve recovery, and to decrease injuries [15,16].

One of the most popular training method implemented in football at all ages and levels of play are the small-sided games (SSGs) [17]. The available literature in football has been related to training routines and drills during the last decades, considering small-sided games (SSGs) as one of the key areas of research. This approach is a training method developed to improve technical, tactical, and positional performances, along with physical attributes, which are the implicit competition and match demands [18]. In fact, there have been many studies carried out in order to determine the different physical [19,20], physiological [21,22], technical [23,24], and tactical [8,25,26,27] demands according to the SSG format and task constraints.

## 2. Research and Development

Based on this rationale, several studies examined the differences between the different SSG formats and analyzed the constraints susceptible of modification by coaches and researchers, shedding light on its complexity [18,28,29,30]. There are many possibilities when reviewing the available literature about the constraints used, such as changing rules, number of players, the use of goals or not, using minigoals or regular goals, encouragement of the coach, or pitch size [18]. Specifically, the pitch size constraint is one of the most relevant when designing SSGs, and has been widely used in previous studies [31], resulting in many game formats (e.g., small and narrow, small and wide, large and narrow, and large and wide [32]. For example, when comparing three different pitch sizes, it was determined that the larger the pitch the higher could the intensity increase and vice-versa [33]. In this sense, it was found that the players covered more distance at high-intensity (HI), when the pitch was larger, and HI deceleration distance was lower, when the pitch had a medium size, in comparison with the larger and smaller ones [34]. Besides that, the impact that the pitch size constraint has in the physical aspect of the players, manipulating pitch size of SSG can be a useful tool to design physical and technical-tactical training tasks based on SSGs [35].

Even though much research on SSGs has been performed during the last years, the vast majority of the studies conducted have focused on field players [18]. To the best of our knowledge, only one study has been carried out comparing the effect of the pitch size on the goalkeeper as a main subject [36]. In this study, Jara et al. [36] aimed to determine how the size of the pitch affected technical and tactical actions of the goalkeeper when playing in three different SSGs. Three different scenarios of SSG were designed and two teams of five-a-side players plus goalkeeper played three matches of 8 min per scenario. They demonstrated that goalkeepers are involved in more defensive and offensive actions in small SSGs. They claimed that varying the size of SSGs can be used as a global training method for the goalkeeper and it would be important to review and analyze how the different variables that define SSGs can modify the goalkeeper’s behavior, in order to design complete and effective training environments for the goalkeeper. One limitation of this study was the lack of analysis from a physical and physiological perspective. As they mentioned, describing others variables such as physiological, physical or tactical must still be the aim of the study to understand the different possibilities that the SSGs can offer to the global training of the goalkeeper. Under this same point of view, it is worth replicating the three scenarios to understand how the different pitch sizes can affect the goalkeeper’s physical and tactical variables to clarify which SSG is adequate in each situation regarding the goalkeeper’s training.

Sainz de Baranda et al. [37] indicated that it is necessary to have a global understanding of the stimulus imposed on goalkeepers during SSGs to optimize the training adaptation. As a global training method, the SSGs allow the integrated training of the technical, tactical, physical, physiological, and psychological aspects not only for the field player, but also for the goalkeeper. Therefore, the quantification of training load is crucial in order to understand the training process and it seems necessary to analyze how SSGs affect the performance of football players in the short and long term. Given the lack of research on goalkeeper performance during SSGs, the aim of the present study was to analyze how the modification of the pitch size in SSGs affects in both tactical and physical demands of the goalkeepers. Because many authors showed for small size SSGs that a higher number of passes, receives, and shots are performed [6,18,33] and that goalkeepers display a higher number of defensive and offensive actions, it was hypothesized that the physical and tactical needs of the goalkeeper in SSGs will be modified when using different pitch sizes. Specifically, goalkeepers should cover larger distances in small SSGs than in medium and large size SSGs. Also, tactical variables are hypothesized to be higher when the pitch size is larger. The following study will help us to understand if there is any difference between such scenarios and will offer new possibilities in task designing.

## 3. Materials and Methods

### 3.1. Participants

Three professional male football goalkeepers participated in this study (age: 24.5 ± 7.2; playing experience: 11 ± 7.9; height (cm): 187.3 ± 2.1; body mass (kg): 80.6 ± 5.5). Ten football field players participated in the protocol, but their data was not analysed. All participants were part of the same team competing in the 5th league. This team was the second team of a professional club playing in the 2. Bundesliga in Germany. The frequency of football practice sessions ranged from 6–8 times a week, 75 min per session (including task focused on the team tactical principles and physical-conditioning training), with an official match during the weekend. All players and technical staff were informed about the research and its procedures, requirements, benefits, and risks. Written consent of all participants (field players and goalkeepers) was obtained before the study began and this investigation was approved by the Institutional Research Ethics Committee of the University and following the recommendations of the Declaration of Helsinki (ID 1944/2018). The coaching staff was informed at all times of all the procedures carried out.

### 3.2. Experimental Task

The teams were composed of six players (5-a-side plus goalkeeper). Field players were divided into two teams (Team A and Team B) taking into account their usual playing position (defender, midfielder, and forward) and their tactical/technical level. To avoid any imbalance between teams, the head coach assigned the players to two balanced teams, adhering to the specific requirements previously used by Casamichana and Castellano [33]. Each goalkeeper played two matches in every SSG scenario. Each goalkeeper player once with each team pro scenario.

Three different small-sided games (SSG) were used in this study, which were similar to previous studies [33,36]. The format of the SSGs were (1) 62 m × 44 m for a large pitch (SSG Large), with a playing area of 2728 m^2^ and a ratio per player of 272.8 m; (2) 50 m × 35 m for a medium pitch (SSG Medium), with a playing area of 1750 m^2^ and a ratio per player of 175 m^2^; and (3) 32 m × 23 m for a small pitch (SSG Small), with a playing area of 736 m^2^ and a ratio per player of 73.6 m^2^ (Figure 1). The individual playing area did not take goalkeepers into account. No specific penalty area was specified in any of the 3 scenarios. The goals used were standard goals use in competition (7.32 m × 2.44 m). The players had to play in the designated area. The rules included some restrictions: there were no corners, thus, every time the ball went out, the goalkeeper of the team with possession put the ball back in play. There were no maximum ball-touches per player. There was no shortage of balls. There were no tactical/technical orders by coaches during the game. There were three matches played of 8 min each in every SSG. A 5-min passive break was taken between matches. The players were allowed to drink water during the break time.

### 3.3. Procedures

This study was conducted over a 3-week period in May during the 2018–2019 season. Each week was dedicated to one type of SSGs. All field players were part of the same team line-up and played on Wednesdays after a standard 20-min warm-up. The warm-up consisted in low intensity running with specific joint mobility exercises and ball possession exercises for the field players or specific technical exercises for the goalkeepers. There were no stoppages for injury. The temperature was similar in all sessions (11.6° ± 4°) considering the influence of climatic variables and its influence on fatigue [38]. To control the learning effects, 9 weeks before leading up to the study all players wore the GPS devices and played in the SSG scenarios used in this study, to familiarise the participants with the various SSG design and the GPS technologies. The GPS devices were activated 15 min prior to training session in order to allow for satellite lock in accordance with previous studies [39] to calculate the location with the higher quality level [40].

#### 3.3.1. Pitch-Positioning Derived-Variables

Positional data of each goalkeeper was gathered using an 18.18 Hz global positioning system (GPEXE GK, Exelio SRL, Udine, Italy). This device was specifically developed by the company for the role of the goalkeeper. The devices were place on the back of each goalkeeper, wearing a small chest vest provided by the company. Latitude and longitude coordinates provided by the device were exported from each unit and transposed into meters using the procedures suggested by Folgado et al. [41]. All data were smoothed using a 3 Hz Butterworth low pass filter and a rotation matrix was calculated for each set of data and computed using dedicated routines in Matlab R2014b Software (The MathWorks Inc., Natick, MA, USA). The overall distance covered and the distance covered at different movement speeds categories were measured as a physical variable of performance. Following the categories used previously in the literature [7,41], the categories were: walking (0.0–3.5 km/h); jogging (3.6–14.3 km/h); running (14.4–19.8 km/h); and sprinting (>19.9 km/h).

#### 3.3.2. Spatial Exploration Index (SEI)

The SEI algorithm proposed by Gonçalves et al. [1] was obtained for each data set of each goalkeeper. This algorithm is calculated by the mean pitch position, computing the distance differences from each positioning time-series to the mean position and computing the mean value from all the obtained distances.

#### 3.3.3. Ellipses

Ellipses are geometrically figures that are used as a spatial analysis for a set of points in a two-dimensional space, which boundaries will enclose about the 100(1 − α)% of the observations or set of points. Ellipses have been used in literature before as tool for spatial analysis in posturography [42] and cartography [43,44].

#### 3.3.4. Standard Ellipse Area (SEA)

A standard ellipse is considered a descriptive tool by Batschelet [45]. It is a classical statistical measure for two dimensional space (*x* and *y*), which represents the dispersion of the bivariate features around its center. Proposed first by Lefever [46], it depicts the spatial distribution of the features and offers a summary of the dispersion and orientation of the data sample [2]. Thus, this ellipse can offer a representation of the data based on the intrinsic properties, showing the average location, dispersion, and orientation of the data analysed [47]. In the standard ellipse, the probability that a given point falls into it is equal to 0.39347 [45]. It is a single measure of the dispersion of the movements of the goalkeeper around the mean center (mean *x*; mean *y*). Thus, the area of the standard ellipse could offer a view of inference zone in a determined field of the goalkeeper; in this case, when analysing the coordinates of the distance covered. This could be an interesting tool in order to analyse the area where the goalkeeper interacted within the task and offering a better understanding of how the goalkeeper is related with the football pitch.

To calculate the ellipse, it is necessary to obtain the mean, minor, and major axes, and orientation of the data set. The mean of the data will become the origin of the ellipses’ axes (1):(1)$$\bar{x}=\frac{1}{n}\overset{n}{\underset{i=1}{\sum }}x_{i},\;\bar{y}=\;\frac{1}{n}\;\overset{n}{\underset{i=1}{\sum }}y_{i}\;$$

Now, the shape and size is obtained from the covariance matrix (Σ) of the data (2) and then the eigenvalues and eigenvectors of the covariance give the lengths of the semi-major axis (*a*) and the semi minor axis (*b*) (3):(2)$$\Sigma \left(\begin{array}{c} var\left(x\right)\;cov\left(x,y\right)\\ \mathrm{cov}\left(y,x\right)\;\mathrm{var}\left(y\right) \end{array}\right)$$
(3)$$a=\lambda_{1}-1\;\;\;\;b=\lambda_{2}-1$$

To obtain the orientation of the ellipse, that will be necessary to illustrate the ellipse, it is calculated the angle (α) of the largest eigenvector towards the *x*-axis as follows (4):(4)$$\alpha =arctan\frac{\nu_{1}\left(y\right)}{\nu_{1}\left(x\right)}$$

From geometry is it known, that the area of an ellipse is the product of the principal axes with the *p* value (5):(5)SEA=πab

#### 3.3.5. Prediction Ellipse Area (PEA)

Batschelet [45] defined the PEA as the ellipse that describes the area in which a single new observation can be expected to fall with a probability estimated from the observed points of the area. Once again, it is essential to get the principal axes of the ellipse. The semi major (*a*) and semi minor axes (*b*) of the prediction ellipse are (6) and (7) [42]:(6)$$a_{p}\sqrt{\frac{2\left(n+1\right)\left(n-1\right)}{n\left(n-2\right)}F\left(1-\alpha \right),2,n-2\cdot \lambda_{1}}\;\approx \;\sqrt{x_{2}^{2}\cdot \lambda_{1}}\;$$
(7)$$b_{p}\sqrt{\frac{2\left(n+1\right)\left(n-1\right)}{n\left(n-2\right)}F\left(1-\alpha \right),2,n-2\cdot \lambda_{2}}\;\approx \;\sqrt{x_{2}^{2}\cdot \lambda_{2}}$$

Just as before, the area of the ellipse is the product of the principal axes (*a_p_* and *b_p_*) with the *p* value (8):(8)$$\mathrm{PEA}=\pi a_{p}b_{p}$$

Thus, this area could offer a view of the inference zone that the goalkeeper could reach based on the distances already covered in a determined field. Figure 2 and Figure 3 show a computed example of both SEA (5) and PEA (8) of a data sample.

### 3.4. Statistical Analysis

Two different statistical analysis were applied to the data. First, the data are presented as mean, standard deviation, and 95% confidence intervals (lower and upper bounds) using a one-way analysis of variance (ANOVA), in order to identify significant differences in each of the dependent variables. The post-hoc Bonferroni test was applied when any significant difference was found between them. The statistical significance was set at *p* < 0.05.

Second, Magnitude-based inferences (MBI) was employed [48]. All processed variables were log-transformed to reduce the non-uniformity of error prior to the scenario comparisons. The comparisons among the different pitch sizes were assessed via standardized mean differences, computed with pooled variance, and respective 90% confidence intervals [49]. Differences in means for both pairs of SSG were also expressed in standardized (Cohen) with 90% confidence limits (CL). The effect was reported as unclear if the CL overlapped the thresholds for smallest worthwhile changes, which were computed from the standardized units multiplied by 0.2. Magnitudes of clear effects were described according to the following scale: 25–75% = possible; 75–95% = likely; 95–99% = very likely; and >99% = most likely [4].

## 4. Results

Table 1 shows the outcomes from the ANOVA and a descriptive analysis of the variables used in the study in the different SSGs. The results when playing SSG Small showed that the distance absolute distance covered increased SSG Medium (mean 123.01; 95% Confidence Intervals(CI) 74.7 to 177.4; *p* < 0.001) and Large (mean 189.9; 95% CI 138.5 to 241.3; *p* < 0.001). The variables SEI, PEA, and SEA increased, when comparing SSG Small with Medium (SEI: mean −1; 95% CI −1.5 to −0.4; *p* = 0.001; PEA: mean −60.4; 95% CI −113.7 to −7.1; *p* = 0.024; SEA: mean −10.1; 95% CI −19 to −1.1; *p* = 0.025) or Large (SEI: mean −0.8; 95% CI −1.3 to −0.2; *p* = 0.009; PEA: mean −61.4; 95% CI −114.7 to −8.1; *p* = 0.022; SEA: mean −10.1; 95% CI −19 to −1.2; *p* = 0.025). There were no significant differences (*p* > 0.05) when analyzing the distance covered in walking intensity between Small and Medium size SSGs. In contrast, significant differences were found when comparing Small with Large (mean 32.3; 95% CI 1.2 to 63.5; *p* = 0.041) or Medium with Large (mean 35.4; 95% CI 4.3 to 66.6; *p* = 0.024) size SSGs. Significant differences were found when comparing the distances covered in jogging intensity of Small SSG with Medium (mean 129.7; 95% CI 69 to 190.4; *p* < 0.001) or Large SSG (mean 157.6; 95% CI 96.9 to 218.3; *p* < 0.001).

Table 2 shows the difference in means using MBI statistical analysis. The SSG Small showed a very likely decrement, when comparing the distance covered with medium SSG. Similarly, when comparing small and medium SSGs with large SSG, the absolute distance covered showed a most likely decrease. In relation with the SEI, medium and large SSG showed a very likely increment in comparison with the small SSG (3.66 ± 0.99 and 3.08 ± 0.81, respectively). In the same way, the results regarding with PEA and SEA showed a most likely increase, when comparing small SSG with medium and large SSGs. Table 2 also shows the results about the distances covered at different intensities. Additionally, the small SSG showed a very/most likely decrement of distance covered at Jogging intensity, when comparing with medium and large SSGs (−2.20 ± 1.84 and −3.62 ± 1.46, respectively). Lastly, when comparing medium SSG with large SSG, a most likely decrement of distance at jogging intensity was observed. Both acceleration and deceleration yielded no significant differences between scenarios.

## 5. Discussion

The present study aimed to identify how the modifications of the pitch size affected the goalkeeper’s physical performance and his/her spatial exploration of the pitch during three types of SSGs (Small, Medium, and Large). It is essential to understand how different training tasks affect the goalkeeper, both in a tactical and physical sense, to provide a suitable training plan as well as an optimal development in this specific role [50]. Analyzing the training sessions is important in order to compare if the demands of the competitions are being properly developed and if the training sessions are giving the players enough tools to perform [10].

In general, there were differences between small and large SSGs in relation to the distances covered at different intensities and pitch exploration. Intensities were lower when the pitch size was larger. Besides that, the pitch exploration variables increased along with the increment of the pitch size. It is remarkable that there were no differences in most of the variables between the SSG medium and SSG large values. Due to the pitch size of the SSG small, the number of shots taken from any position on the field is much higher than in the other ones. If we compare the SSG small with real competition, it can be appreciated that almost any area of this SSG would be in a suitable distances or angle to shot with a high probability of scoring or at least aiming target. As reported by Sainz de Baranda et al. [51], the zones where the majority of the shoots were taken part in competition are the only zones and angles present in the SSG small. For this reason, SSGs medium and large would allow teams to build up the play in a more elaborate way and not aiming to score right after winning the possession of the ball or putting the ball in play [33]. The three different SSGs examined in this study showed clear tendencies concerning the distances covered by the goalkeepers. Distances decreased as the pitch size of the SSGs increased. In line with the expectations, those differences can be explained due to the higher number of defensive and offensive actions that take part in small SSG by the goalkeeper [36]. Many authors showed that a higher volume of passes, receives, and shots are performed in smaller pitches [6,18,33]. Accordingly, more technical actions by the goalkeepers are required, implying a constant physical activity in order to cover those behaviors. By comparing the results obtained with the results in competition, there is a clear difference between matches and SSGs. Some studies [50,52,53] showed that goalkeepers covered more distance at walking intensity than other intensities during a match. From this perspective, the results of the present study are in concordance with the overall results of this study. However, when using small SSGs, the volume of distance covered at jogging intensity were higher than at walking intensity. Although in medium and large SSGs the volume of distance covered at walking intensities are higher than the distance covered at jogging intensity, distance covered at jogging intensity showed high volumes in both pitch sizes. Thus, it may be suggested that SSGs require higher physical intensity than match-derived demands [54]. This may be because of the high number of actions that SSG requires, with the smaller SSGs requiring more actions by the goalkeeper than the larger SSGs [36]. This finding can be seen in the number of accelerations and decelerations that are performed by the goalkeeper across the different SSGs. When comparing the results of the present study with previous ones focused on match-demands [10,50], it can be observed that in 8 min that a SSG match lasted, the number of accelerations and decelerations are relatively higher than the number of accelerations and decelerations performed in 90 min that a match lasts. It has to be kept in mind, however, that the goalkeeper’s activity is intense and it is likely related to technical behaviors, when designing the training sessions.

The analysis of variables related to the pitch exploration revealed some performance trends. The increase of the pitch size of SSGs increased the SEI, SEA, and PEA values. The goalkeepers explored more space when the SSGs were larger. Larger pitch sizes allow the player to reach higher values related to the pitch exploration [55]. All values of the pitch exploration variables are strongly related to the specific role of the goalkeeper who has the primary role of preventing the opposing team from scoring and the pitch size of the SSGs. In fact, these results are most plausible, because the goalkeeper position is a role that is more predictable in his/her position than other roles and stays most of the time in the same part of the field due to his/her tactical duties, showing a low spatial variability [56]. Higher values in all of these variables are related to goalkeepers covering more space during the training or game [55]. Nevertheless, this specific role within football will reach lower values than others roles as long as the tasks are designed within a global methodology. Also, the values of the PEA variable are related to the probabilities of covering a certain space during the next observation. This variable can be used in order to know how a task can be developed regarding the exploration of the space, being an interesting tool in this area [7].

It should be noted that the present study has some limitations, which may have influenced the results. The low number of participants for the analysis does not allow one to overgeneralize the findings of the present study. The lack of the analysis of the interrelationships between physical and technical and tactical actions, as well as the need for increasing the control and modification variables are other limitations. On the other hand, the order of the SSGs was not randomized. Although players were accustomed to this quantity and type of SSGs, fatigue could have affected the players’ responses. To avoid this situation, a recovery of 5 min was included in the study. Previous studies suggest recovery times >4 min do not impact the physical and physiological demands of multiple SSGs [57]. Furthermore, it is necessary to take the specific rules of the SSGs into account, that included some restrictions, such as keeping play in the designated area, no corners, and every time the ball went out, the goalkeeper of the team with possession made a goal kick. Another limitation is the lack of unanimity to establish activity profiles in terms of intensity and the lack of studies in this regards for goalkeepers.

Because of the scarce knowledge about the effects of training load on goalkeeper’s physical performance and on how this performance is affected by the modification of the pitch size of SSGs, the present findings are a first step to expand on the limited data of previous research. It was of importance to highlight that GPS is an emerging technology that can be further utilized in applied research on football goalkeepers. The ability to collect data during training sessions will allow researchers access to unprecedented information.

## 6. Conclusions and Practical Applications

Using pitch area modifications seems to be a key aspect when planning the training drills of goalkeepers, due to the differences in the physical-related variables and the spatial exploration. As shown in this study, distances covered and their intensities are not related to the spatial exploration of the goalkeeper and higher volumes of distances covered at different intensities are not related to higher values of the different spatial explorations variables. Increasing the pitch size of SSGs has a significant impact on how the goalkeeper can reach a higher spatial exploration of the pitch. When aiming to improve tactical concepts in a global context similar to the competition, larger pitch areas may contribute to this approach. Regarding to the physical performance, when aiming to provide a more intense training to goalkeepers, smaller pitch areas can contribute to this goal.

## Figures and Tables

**Figure 1 sensors-19-03605-f001:**
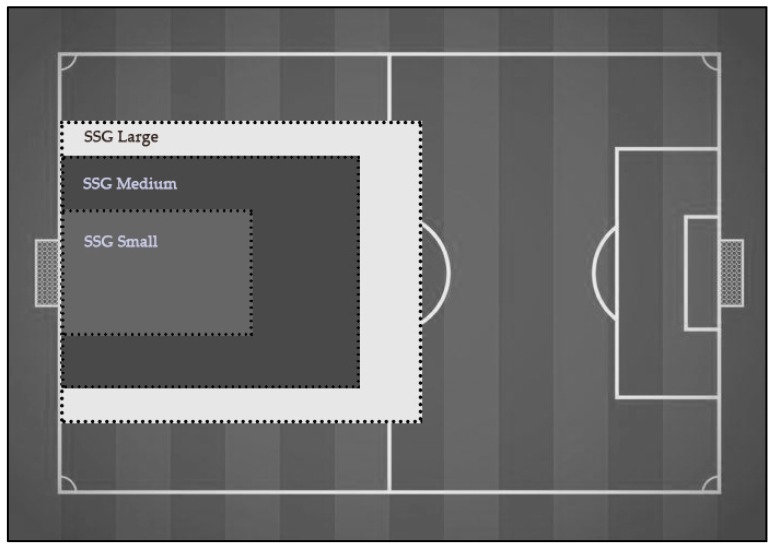
Three different pitch sizes of the SSG compared to the normal pitch size of a football pitch. The individual playing area did not take goalkeepers into account [36].

**Figure 2 sensors-19-03605-f002:**
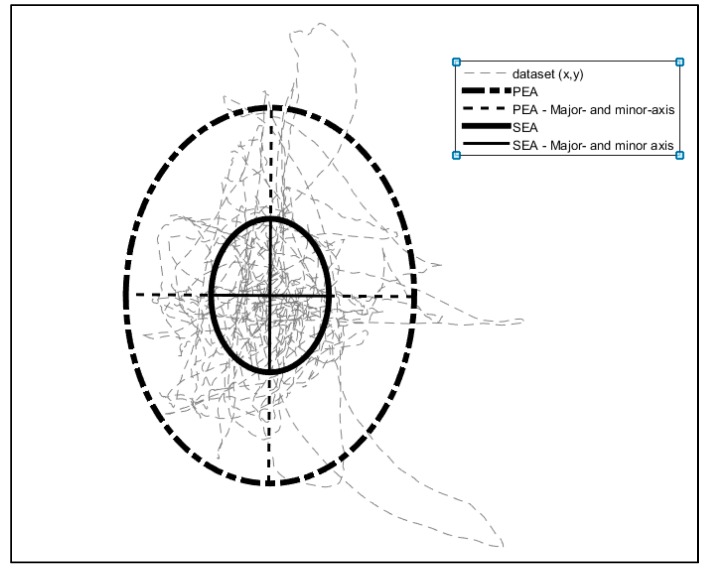
Representation of the PEA and SEA. Major and minor-axis of PEA and SEA are represented. The rotation (α) of both ellipses is established using the formulae 4 and is applied through the *x*-axis.

**Figure 3 sensors-19-03605-f003:**
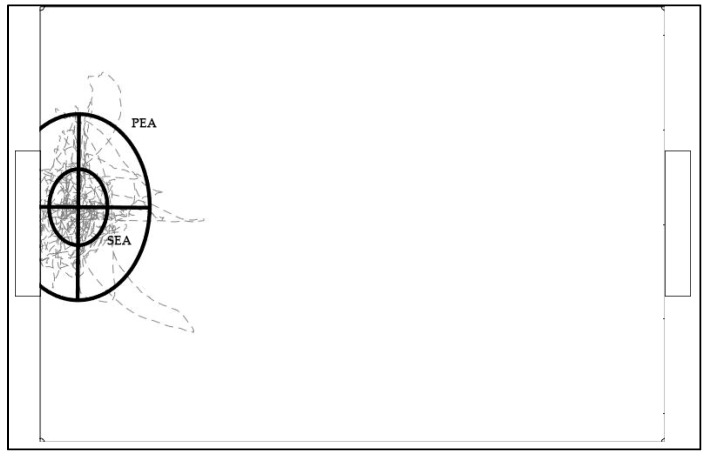
Representation of the SEA and PEA in a SSG small (32 m × 23 m).

**Table 1 sensors-19-03605-t001:** Descriptive analysis (mean ± SD) and 95% confidence interval for mean (in parenthesis lower and upper bound).

Variables	SSG Scenarios		
	Small	Medium	Large
**Absolut Distance Covered (m)**	445.1 ± 44.3 (398.6 to 491.6) ^a,b^	319 ± 25.3 (292.5 to 345.6) ^c^	255.2 ± 25.9 (228.1 to 282.4)
**SEI**	2.2 ± 0.3 (1.9 to 2.5) ^a,b^	3.2 ± 0.4 (2.8 to 3.6)	3 ± 0.4 (2.5 to 3.4)
**PEA (m^2^)**	67.6 ± 18.8 (47.9 to 87.4) ^a,b^	128 ± 31.6 (94.8 to 161.2)	129.1 ± 46.5 (80.2 to 177.9)
**SEA (m^2^)**	11.3 ± 3.1 (8.0 to 14.6) ^a,b^	21.4 ± 5.3 (15.8 to 26.9)	21.4 ± 7.8 (13.2 to 29.6)
**DISTANCE COVERED** **Walking (<3.5 km/h)**	159 ± 12.6 (145.8 to 172.2) ^a^	162.1 ± 25.8 (135.0 to 189.1) ^c^	126.6 ± 19.6 (106.1 to 147.2)
**Jogging (3.6–14.3 km/h)**	278.6 ± 48.6 (227.6 to 329.5) ^a,b^	148.9 ± 42.4 (104.4 to 193.4)	120.9 ± 20.3 (99.6 to 142.2)
**Running (14.4–19.8 km/h)**	7.6 ± 5.3 (2 to 13.2)	5.1 ± 4.6 (0.3 to 9.9)	5.9 ± 4.4 (1.3 to 10.5)
**Sprint (>19.9 km/h)**			1.6 ± 2.1 (−0.6 to 3.8)
**Number of Accelerations (>3 m/s^2^)**	5.5 ± 3.9 (1.4 to 9.6)	2.7 ± 1.9 (0.7 to 4.6)	4 ± 0.6 (3.3 to 4.7)
**Number of Deccelerations (<−3 m/s^2^)**	4.2 ± 2.9 (1.1 to 7.2)	2.5 ± 1.6 (0.8 to 4.2)	4 ± 1.1 (2.9 to 5.1)

Note: Post-hoc Bonferroni test: ^a^ SSG Small vs. SSG Large; ^b^ SSG Small vs. SSG Medium; ^c^ SSG Medium vs. SSG Large (*p* < 0.05 in all cases). SEI, Spatial Exploration Index; PEA, Predictive Ellipse Area; SEA, Standard Ellipse Area.

**Table 2 sensors-19-03605-t002:** Difference in means (Standarized; ±90% CL) and uncertainty in the true differences comparisons among SSG pitch sizes.

Variables	(a) Small v Medium	(b) Small v Large	(c) Medium v Large
Absolute Distance Covered (m)	−2.23 ± 1.2	−4.31 ± 1.59	−2.08 ± 0.63
	very likely ↓	most likely ↓	most likely ↓
SEI	3.66 ± 0.99	3.08 ± 0.81	−0.58 ± 1.47
	most likely ↑	most likely ↑	unclear
PEA (m^2^)	3.15 ± 0.96	3.05 ± 1.27	−0.1 ± 1.54
	most likely ↑	most likely ↑	unclear
SEA (m^2^)	3.15 ± 0.96	3.01 ± 1.28	−0.14 ± 1.57
	most likely ↑	most likely ↑	unclear
Distance Covered Walking (<3.5 km/h)	−0.57 ± 1.86	−2.93 ± 2.09	−2.36 ± 1.58
	unclear	very likely ↓	very likely ↓
Jogging (3.6–14.3 km/h)	−2.20 ± 1.84	−3.62 ± 1.46	−1.42 ± 0.54
	very likely ↓	most likely ↓	most likely ↓
Running (14.4–19.8 km/h)	1.17 ± 2.2	0.27 ± 0.67	−0.64 ± 1.47
	unclear	unclear	unclear
Sprint (>19.9 km/h)	unclear	unclear	unclear
Number of Accelerations (>3 m/s^2^)	−2.9 ± 4.85	−0.33 ± 4.59	2.57 ± 6.95
	unclear	unclear	unclear
Number of Decelerations (≤3 m/s^2^)	−1.91 ± 2.85	−1.36 ± 2.88	0.55 ± 3.87
	unclear	unclear	unclear

Note: CL = confidence limits; ↑ = increase; ↓ = decrease; tri = trivial. Comparisons among SSG scenarios are identified as: (a) SSG Medium vs. SSG Small; (b) SSG Large vs. SSG Small (c) SSG Large vs. SSG Medium. SEI, Spatial Exploration Index; PEA, Predictive Ellipse Area; SEA, Standard Ellipse Area.

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
