# Peer review of "Physical and Tactical Demands of the Goalkeeper in Football in Different Small-Sided Games"

_sensors, 2019, doi:10.3390/s19163605_

Round 1

Reviewer 1 Report

The objective of the manuscript is to analyse how the modification of the pitch size in Small-Sided Games (SSGs) affects the physical demands of 21 the goalkeepers. The study of physical demands of SSG is not new at all as many studies have been conducted since 2008. Indeed, now we know that the inclusion of goalkeepers in SSGs change both the tactical and physical demands of football players.

However, despite these studies revealed the physical demands of different SSGs formats in field players, up to my knowledge, the impact of the modification of SSGs formats in the physical demands of goalkeepers is unknown. Therefore, I find this manuscript relevant. Nonetheless, there are important issues that must be addressed before considering it for publication

The most important issues are:

1.       Methodological issues: The methods of this study are incomplete, there is important information missing like the size of penalty areas in each SSGs format, the number of SSGs played by each goalkeeper, the goalpost size, etc.

2.       Results: The first paragraph has important errors to address. Moreover, Table 1 also contains typos 

3.       Figures: are not appropriately designed. First, it is recommended to avoid colours when possible. Second, the lines are very tight to recognise some colours. Third, figure 2 and 3 should be bigger. Fourth, goalpost and penalty area are not represented in figures 1 and 3. Fifth, in figure 1 there are some black lines that should be removed. Etc… 

4.       English Speaking: need to be professional proofread. It mixes the American style with the British style. Also, some paragraphs are difficult to understand and there are some grammatical issues

Please, see the attached file to get further feedback.

Author Response

please find responses in the attachment.

Reviewer 2 Report

The study presented analyzes how the modification of the pitch size in SSGs affects the physical demands of the goalkeepers. In this sense, it is a novel study both for the material used and for the study population.

In the introduction, the authors indicate that the use of GPS can help to reduce injuries and it would be interesting that some author contributed to support this issue (Line 49). For this, one of the following studies is recommended:

Robles-Palazón, F. J.; Cejudo, A.; Ayala, F.; Sainz de Baranda, P. (2019). Características de las estrategias de prevención de lesiones en niños y adolescentes deportistas. Revisión sistemática. Journal of Sport and Health Research. 11(1):1-16.

Olmedilla, A.; García-Alarcón, M.; Ortega, E. (2018). Relaciones entre lesiones deportivas y estrés en fútbol 11 y fútbol sala femenino. Journal of Sport and Health Research. 10(3):339-348.

In the experimental task, the authors indicate that the Casamichana and Castellano criteria for the formation of balanced teams were followed. This variable is very important and determines the development of the tasks, so it would be interesting for the authors to clearly add these criteria.

Finally, although it is not a limitation of the study, the authors could consider external variables such as climatic ones since they can directly affect the fatigue of the goalkeepers and that can cause the results of this study to be different although the subjects were the same In this way, the following study can help the authors to extract ideas in this regard:

Gutiérrez, J.; Casamichana, D.; Castellano, J.; Sanchez-Sanchez, J. (2018). Influencia de la localización geográfica de los partidos de fútbol en la respuesta física de equipos  que compiten en la Segunda División Española. Journal of Sport and Health Research. 10(2):295-302.

In the section "practical applications" it is necessary to appear the word "conclusions" since in that section the objective of the work is answered based on the results found and although the section can be named for the practical applications of this work, It should be indicated which is the main conclusion that responds to the objective.

Correct the misspelling on line 314 “inciación”

Author Response

Response to Reviewer 2 Comments

Point 1: The study presented analyzes how the modification of the pitch size in SSGs affects the physical demands of the goalkeepers. In this sense, it is a novel study both for the material used and for the study population.

 In the introduction, the authors indicate that the use of GPS can help to reduce injuries and it would be interesting that some author contributed to support this issue (Line 49). For this, one of the following studies is recommended:

 Robles-Palazón, F. J.; Cejudo, A.; Ayala, F.; Sainz de Baranda, P. (2019). Características de las estrategias de prevención de lesiones en niños y adolescentes deportistas. Revisión sistemática. Journal of Sport and Health Research. 11(1):1-16.

Olmedilla, A.; García-Alarcón, M.; Ortega, E. (2018). Relaciones entre lesiones deportivas y estrés en fútbol 11 y fútbol sala femenino. Journal of Sport and Health Research. 10(3):339-348.

Response 1: First of all, we would like to thank you about the believe of the novel contribution of this specific manuscript in this specific topic.

The studies you mentioned have been added and updated according to your advise.

Point 2: In the experimental task, the authors indicate that the Casamichana and Castellano criteria for the formation of balanced teams were followed. This variable is very important and determines the development of the tasks, so it would be interesting for the authors to clearly add these criteria.

Response 2: We appreciate this suggestion and we improved the description of the teams and how they were built.

Point 3: Finally, although it is not a limitation of the study, the authors could consider external variables such as climatic ones since they can directly affect the fatigue of the goalkeepers and that can cause the results of this study to be different although the subjects were the same In this way, the following study can help the authors to extract ideas in this regard:

 Gutiérrez, J.; Casamichana, D.; Castellano, J.; Sanchez-Sanchez, J. (2018). Influencia de la localización geográfica de los partidos de fútbol en la respuesta física de equipos  que compiten en la Segunda División Española. Journal of Sport and Health Research. 10(2):295-302.

Response 3: We appreciate the suggestions regarding the climatic inference. We introduced this recommendation in the Procedures section. We exposed that the sessions were performed always under the weather conditions to avoid this problem.

Point 4: in the section "practical applications" it is necessary to appear the word "conclusions" since in that section the objective of the work is answered based on the results found and although the section can be named for the practical applications of this work, it should be indicated which is the main conclusion that responds to the objective

 Response 4: We thank the reviewer for this appreciation. The section was renamed. The main conclusion as previously mentioned is: “When aiming to improve tactical concepts in a global context similar to the competition, larger pitch areas may contribute to this approach. Regarding to the physical performance, when aiming to provide a more intense training to goalkeepers, smaller pitch areas can contribute to this goal”.

Point 5: Correct the misspelling on line 314 “inciación”

 Response 5: The misspelling was revised. Thanks for this appreciation.

Round 2

Reviewer 1 Report

Despite the study of the physical and tactical demands of small-sided games is not new, up to my knowledge, the impact of the modification of SSGs formats in the physical demands of goalkeepers is unknown. This manuscript provides an initial scope about this regard and their findings should be considered by both researchers and trainers.

Regarding this second version, I congratulate the authors because the manuscript has been widely improved and have successfully addressed all the recommendation provided by the Reviewer 1 and Reviewer 2. All in all, there are still some minor changes that can be addressed to increase the quality of the work.

INTRODUCTION

I don't understand why the authors have divided the introduction into two parts 1) State of art, and 2) Research and Development. I think it looks much cooler in the first version. If possible, I highly recommend the authors to go back to the initial proposal and make a general introduction that shows the gap in literature address by this research.

Paragraph 1. Lines 40-44: I think these two sentences can be merged together. Moreover, they can be better written to better show the authors’ statement. Also, I think the first sentence has some relevant grammar mistakes. In any case, Please, add the corresponding reference to support this statement Paragraph 1. Lines 44-50: these sentences somehow repeat the same stated before. I think the authors have to rewrite the paragraph 1 and go directly to the point they want to state, which actually is the importance of the use of GPS technology in training and the different ways to use this technology in training.

METHODS

Participants, Line 114-115: The sentence is not well written and can mislead the reader. I know the sentence is quite tricky but, please rewrite. Readers must clearly understand that players belonged to a team playing in the 5th division in Germany. However, this team is the second team of a professional club whose first team is playing in The 2. Bundesliga.

Experimental task, Line 136 “The individual playing area did not take goalkeepers into account“: Because the playing area is already specified in line 134. This sentence should be moved above. Actually, I encourage the authors to add this information in brackets right before “(Figure 1)”

Experimental task, Line 139-140: there is the following typo “The coaches”. These two words are between two different sentences so must be a mistake, please delete.

Experimental task, Line 142: Please specify the drinks that were allowed to drink during breaks.

Procedures. Line 152: Please add the mean and standard deviation for the temperature and humidity.

RESULTS

Line 262: Please avoid p < 0.000. Instead write p < 0.001

DISCUSSION

Lines 344-346: Rewrite the sentence because contain some important grammar mistakes. I recommend the authors to go straight to the point. “Another limitation is the lack of unanimity to establish activity profiles in terms of intensity and the lack of studies in this regards for goalkeepers.

CONCLUSIONS

Line 354: avoid writing “the coaches’ perspective”. The authors have to be confident in their work because the conclusions of this study are highly supported by the outcomes collected, although much research is required

Lines 354-359: I recommend the authors to go straight to the point. Avoid the use of connectors like “however”. Conclusions are supported by the main outcomes of the study so just state the main conclusions and if the appropriate practical application of these conclusions.

Author Response

Response to Reviewer 1 Comments

Despite the study of the physical and tactical demands of small-sided games is not new, up to my knowledge, the impact of the modification of SSGs formats in the physical demands of goalkeepers is unknown. This manuscript provides an initial scope about this regard and their findings should be considered by both researchers and trainers.

Regarding this second version, I congratulate the authors because the manuscript has been widely improved and have successfully addressed all the recommendation provided by the Reviewer 1 and Reviewer 2. All in all, there are still some minor changes that can be addressed to increase the quality of the work.

First of all, we would like to thank you about your words about our manuscript. We want to thank you about this reviewing process because it helped us to improve our study beyond the mere publishing process. We addressed the minor changes required in detail.

Point 1: I don't understand why the authors have divided the introduction into two parts 1) State of art, and 2) Research and Development. I think it looks much cooler in the first version. If possible, I highly recommend the authors to go back to the initial proposal and make a general introduction that shows the gap in literature address by this research.

Response 1: We divided this part into two because the academic editor suggested it. “Authors should divide section 2 in two sections. First, State of the Art, Second, Research and Development. In the actual state both parts are mixed into section 2.”

Point 2: Paragraph 1. Lines 40-44: I think these two sentences can be merged together. Moreover, they can be better written to better show the authors’ statement. Also, I think the first sentence has some relevant grammar mistakes. In any case, Please, add the corresponding reference to support this statement Paragraph 1. Lines 44-50: these sentences somehow repeat the same stated before. I think the authors have to rewrite the paragraph 1 and go directly to the point they want to state, which actually is the importance of the use of GPS technology in training and the different ways to use this technology in training.

Response 2: As stated above, the academic editor suggested us to divided into two parts. That is why we understood that a long introduction to the GPS technology in research could help us to a better development of this section. Anyway, we merged the first suggestion and erased the repeated sentence.

Point 3: METHODS

Participants, Line 114-115: The sentence is not well written and can mislead the reader. I know the sentence is quite tricky but, please rewrite. Readers must clearly understand that players belonged to a team playing in the 5th division in Germany. However, this team is the second team of a professional club whose first team is playing in The 2. Bundesliga.

Response 3: We tried to rewrite this sentence in order to clear it. We thank you for this.

Point 4: Experimental task, Line 136 “The individual playing area did not take goalkeepers into account“: Because the playing area is already specified in line 134. This sentence should be moved above. Actually, I encourage the authors to add this information in brackets right before “(Figure 1)”

Response 4: We modified the suggestions proposed.

Point 5: Experimental task, Line 139-140: there is the following typo “The coaches”. These two words are between two different sentences so must be a mistake, please delete.

Response 5: We corrected the typo. Thank you for this appreciation.

Point 6: Experimental task, Line 142: Please specify the drinks that were allowed to drink during breaks.

Response 6: We added what the players drank during the present manuscript.

Point 7: Procedures. Line 152: Please add the mean and standard deviation for the temperature and humidity.

 Response 7: Temperature was added to the procedures. Humidity was not measured.

Point 8: RESULTS

Line 262: Please avoid p < 0.000. Instead write p < 0.001

 Response 8: We modified the p. Thank you for the suggestion.

Point 9: DISCUSSION

Lines 344-346: Rewrite the sentence because contain some important grammar mistakes. I recommend the authors to go straight to the point. “Another limitation is the lack of unanimity to establish activity profiles in terms of intensity and the lack of studies in this regards for goalkeepers.

 Response 9: This sentence was modified taking in account your suggestions. We thank you for this.

Point 10: CONCLUSIONS

Line 354: avoid writing “the coaches’ perspective”. The authors have to be confident in their work because the conclusions of this study are highly supported by the outcomes collected, although much research is required

Response 10: Conclusions were modified as required.

Point 11: Lines 354-359: I recommend the authors to go straight to the point. Avoid the use of connectors like “however”. Conclusions are supported by the main outcomes of the study so just state the main conclusions and if the appropriate practical application of these conclusions.

Response 11: We tried to go straight to the point in the conclusions. That is why all connectors were suppressed. Thanks for this appreciation.

Round 3

Reviewer 1 Report

I congratulate the authors the opportunity of reading their work. Also, I want to congratulate them as they have effectively addressed all the proposed suggestions. I have no further suggestion to add.